# The deciduous genital spines of the moth *Peridea anceps* (Goeze, 1781): Potential socially transferred materials

Saúl Bernat-Ponce[1], Jose Vicente Pérez Santa Rita [1], Carlos Cordero [2*], Joaquín Baixeras [1*]

1 Instituto Cavanilles de Biodiversidad y Biología Evolutiva, Universidad de Valencia, Valencia, Spain,
2 Instituto de Ecología, Universidad Nacional Autónoma de México, Mexico City, México

* cordero@ecologia.unam.mx (CC); joaquin.baixeras@uv.es (JB)

## Abstract

The genitalia of Lepidoptera are complex structures that evolve rapidly and divergently. The endophallus of many lepidopterans is ornamented with elaborate sclerotized structures known as cornuti. In some species, the cornuti are deciduous and remain within the female genital tract after copulation; the function of these structures is virtually unknown. These structures are a peculiar potential type of secondary "socially transferred material" (the primary materials are spermatozoids, i.e., genetic material), because they probably influence the physiology and behaviour of receivers (i.e., females) but are not molecules that function as allohormones. Such influence could be achieved by acting as mating plugs or via mechanical stimulation of females. The most intriguing and bizarre deciduous cornuti are the so-called caltrop cornuti, star-shaped structures, composed of several rays radiating from a central mass. Despite the presence of caltrop cornuti in at least 400 species, there are no studies of their microscopic structure and mechanism of dislodgement and transference to the female. In this study, we describe in detail the general and microscopic structure of the caltrop cornuti and associated structures of the prominent moth *Peridea anceps* (Notodontidae). We provide quantitative data of the cornuti and, for the first time, we explain the processes of detachment and transference to the female genital tract; we also describe their distribution inside the female genitalia. We discuss the possibility that one of the functions of deciduous caltrop cornuti is protection against sperm competition, as well as their potential influence on other aspects of the behaviour and physiology of females via mechanical stimulation.

## Introduction

Genitalia are among the most diverse organs of the anatomy in Lepidoptera [1]. Their complex structure has evolved rapidly and divergently [2–5], although there

**Data availability statement:** All relevant data are within the paper and its Supporting Information files.

**Funding:** Saúl Bernat was supported by a PhD grant of the Ministerio de Ciencia, Innovación y Universidades (MICIU) of the Spanish Government. Carlos Cordero was supported by a sabbatical grant from PASPA/DGAPA (Universidad Nacional Autónoma de México) and by a grant from the University of Valencia (subprogram "Atracció de Talent"), Spain.

**Competing interests:** The authors have declared that no competing interests exist.

are exceptions in some groups. Although their use in taxonomic practice is old and extensive, the interpretation of the evolutionary forces involved in its configuration is still incomplete [6]. The male intromittent organ, the phallic tube, is usually a rather sclerotized simple structure. However, inside the phallic tube there is a mostly membranous endophallus (taxonomically vesica), often complex, that is everted into the female bursa copulatrix and carries the genital pore. The bursa copulatrix is normally a bag like structure that receives and, generally, process (digest) the male spermatophore and other substances [7]. The endophallus is often ornamented by elaborate sclerotized structures, often teeth or plate like, collectively known as cornuti [1]. Number, disposition, and shape of the cornuti are taxon dependent [1,6]. Several functional hypotheses, not always incompatible, have been proposed for the cornuti [8,9].

Among the most intriguing cornuti are those deciduous that generally break off during the copula and remain inside the female genital tract [1,8–11]. There are scattered records of deciduous cornuti in several families of Lepidoptera [8,9] and they are well known in the family Tortricidae, where they constitute a well-characterized common element [10,12].

A recent conceptual synthesis [13] suggests that the effects on the behaviour, physiology and fitness of the diverse materials transferred between conspecifics (e.g., ejaculates, milk, etc.) share traits in common, including their influence in the evolution of many animal traits via indirect genetic effects. These "socially transferred materials" are materials produced by the donor that directly influence the physiology of the receiver bypassing its sensory organs and provide a benefit the donor [9]. We think that deciduous cornuti will be shown to be secondary socially transferred materials (the primary materials being spermatozoids, i.e., genetic material) once the hypotheses on their effects on the physiology and behaviour of females [8] are tested.

Among the most intriguing and bizarre morphologies, the so-called caltrop cornuti stand out. First described by Rothschild and Jordan [14] and so named after Chapman [15], they are star-shaped structures, composed of several rays radiating from a central mass. Caltrop cornuti have been reported in more than 400 species mostly belonging to the Notodontidae and a few species in the Sphingidae, Nolidae and Geometridae [9]. In the Notodontidae their presence is widely extended in the subfamilies Platychasmatinae, Heterocampinae, Phalerinae, Dudusinae, Notodontinae, Nystaleinae and Dioptinae. These cornuti are known to be variable in number, size, and position [9,16,17], although no detailed analysis of their microscopic structure as well as their mechanism of dislodgement and transference has been proposed. In this article we study in detail the microscopic functional structure and variability of the caltrop cornuti (from now on, cornuti) in the prominent moth *Peridea anceps* (Goeze, 1781).

## Methods

The adults of *P. anceps* examined in this study include museum specimens as well as live specimens (collected or reared) fixed for microscopic study. A total of 65 dry specimens (18 females, 47 males) from entomological collections including material collected by the authors, were examined. Another set of 48 specimens (1 female, 47 males) was collected in 2017–2023 in locations of the Valencian Community (Spain),

especially in Chelva (Valencia). Tower traps equipped with 12W actinic light fluorescent tubes were used (Bioform ©). Finally, 54 siblings (31 males, 23 females) were obtained by captive breeding of the eggs laid by a gravid female collected in the field in May of 2019. S1 Table gives details of all the specimens used in this work. Some specimens of other not-odontid species were examined for comparison: *Harpyia milhauseri* (Fabricius, 1775), *Drymonia querna* (Denis & Schiffermüller, 1775) and *Drymonia dodonaea* (Denis & Schiffermüller, 1775).

A few live specimens were treated with dichlorvos (2,2-Dichlorovinyl dimethyl phosphate) for in situ observation of vesica eversion and ejaculation following a similar approach to Dang [18] and Zlatkov [19]. Field male specimens were placed in large glass tubes (approximately 250 ml) that had previously contained for few seconds a solid plastic pellet impregnated with dichlorvos. Specimens were placed in the tubes but removed as soon as they showed the slightest sign of being affected by the insecticide. Using adhesive tape, they were attached by the wings to glass plates, in an inverted position, so that we could film ventrally the abdominal tip under the stereomicroscope.

Dry specimens allow dissection of the integument and removal of the cornuti but cannot be used to study soft tissues. Collected specimens intended for simple dissection were preserved in 70° ethanol. Some specimens were fixed and preserved in a mixture of ethanol 96° and formaldehyde 40% (1:2). Finally, specimens for electron microscopy were fixed with Karnovsky's fixative [20] in phosphate buffer (with addition of 0.3% Triton). The fixative was injected in specimens anesthetized with ethyl acetate. In these cases, dissection was performed immediately (see below) and a new fixation was applied on the dissected and sectioned phallic tube to facilitate penetration of the fixative. In some specimens the phallic tube was fixed during eversion following Zlatkov et al. [21].

The dissection procedure for museum specimens was inspired in protocols by Robinson [22] and Dang [18], meanwhile the dissection of freshly ethanol fixed specimens followed Zlatkov et al. [21]. Dissections were performed in embryo dishes under a Leica MZ9.5 stereomicroscope.

The cornuti can be examined in detail only by removing them from the phallus. In a KOH digested and everted phallus, a careful incision with spring scissors allowed access to the cornuti. Using a sharp tungsten needle, the set of cornuti was removed from the tissue and gently slipped on a small water drop on a microscope slide. The set of cornuti was then disaggregated and conveniently displayed on the microscope slide. The slide was then dried at room temperature. One drop of absolute ethanol ensured dehydration and fixed the position of the cornuti. Finally, a small drop of Euparal mounting media was added to the preparation that was gently covered by a cover slip.

In females, the KOH digestion was reduced as much as possible to allow observation of the position of the cornuti in relation to the bursa copulatrix and remains of the spermatophore. Unfortunately, the process that the female genitalia undergo (digestion by KOH, dehydration, clearing) also allows the cornuti to move inside the bursa copulatrix, and consequently there may be some artifact in these observations.

To obtain better results in the photographs of genital structures through optical microscope, the structures were cleared. Male phalli and female bursae copulatrix were dehydrated in ethanol of increasing gradient up to absolute ethanol. The genitalia or fragments were then immersed in 98% methyl salicylate until the desired point of clearing was achieved.

The cornuti were also studied by scanning electron microscopy. The cornuti were prepared both from male phalli and from female bursa copulatrix. For scanning electron microscopy (SEM) observation, we proceeded with the dissection of the phallus in a similar way to light microscopy observation. Critical point was used to dry the fragments in microporous cylinders. Cylinders with dry cornuti were overturned on 15 mm Hitachi stubs (Ted Pella© 16084−20) previously covered with carbon adhesive tape (G3347N Agar Scientific). The cornuti were then lightly pressed on the adhesive surface with the help of a tungsten tip to ensure their fixation. When cornuti were to be observed adhered to the inner surface of the female bursa copulatrix, a fragment of integument was cut out of the previously dissected bursa and adhered to the corresponding stub in a similar manner as was done with the isolated cornuti. The preparations were sputter coated with AuPd using a Sputter Coater Polaron SC7640 for 100 seconds (approximately at a speed of 3 Å/s). The samples thus prepared were observed and photographed using a Hitachi S-4800 scanning electron microscope.

For transmission electron microscopy, after the first fixation, the phallic tube was cleanly dissected and cut into two fragments to facilitate penetration of the fixative. The fragments thus obtained were washed with phosphate buffer and post-fixed with 2% osmium tetroxide. Fragments were washed, dehydrated in increasing ethanol concentration, and embedded in Durcupan resin (Fluka). Semi-thin sections (1.5 μm) were cut with a diamond blade on a Leica EM UC6 ultratome and stained with 1% toluidine blue. These sections were observed and photographed with an optical microscope (Leica DMLB see below). Ultrathin sections (0.08 μm) were then obtained and stained with lead citrate. The slides were examined on a Hitachi HT7800 transmission electron microscope.

A Leica Z16 and Leica DMLB microscopes were used for photo documentation and observation, both equipped with a Leica CF500 camera and LAS 4.13 software. Both the number of cornuti and the number of rays of every star-shaped cornutus were counted with the help of printed photographs and direct observation of the preparations through the microscope. The count in each preparation was done twice and a third time in case of disagreement.

To consider variables potentially correlated to the number of cornuti and number of rays, we weighed the whole body and measured wing length, sclerotized phallic tube length and right valva length of male moths (S1 Fig). Body weight and wing length were measured in some of the adults collected by the authors in 2022 and some of the museum specimens. The collected specimens were spread out and left to dry for one week and then weighed. All weights were obtained in a Mettler Toledo SM635i-ION scale. Wing length was measured from the base of the wing to the very apex including fringe [23]. Due to the special restrictions imposed by the COVID-19 pandemic, the group of siblings emerged in the laboratory did not receive adequate care. Consequently, their wings, seriously damaged, were inadequate for measuring. However, length of the valva and length of the sclerotized phallic tube were measured in all male siblings with an eyepiece micrometer.

Statistics of the number of cornuti ($\bar{X}$, SD, min-max, N) for the total specimens, the whole group of siblings, field specimens and museum specimens were calculated. Statistics of the number of rays per specimen and per cornutus for the whole group of siblings, some field specimens and some museum specimens were also obtained.

All statistical analyses were implemented in the "stats" package in R Studio [24]. Prior to each analysis, a Shapiro-Wilks test ("shapiro.test") was performed to check the normality of the variables. Variance Inflation Factor was evaluated through "vif" function in the "car" package [25]. For the group of male siblings, a General Linear Model (GLM) was used to evaluate the possible correlation between genital size, estimated as the sclerotized phallic tube length (PL1, PL2) and right valva length (RVL), and the number of cornuti. Pearson and Spearman correlations were performed with the "cor.test()" function depending on the normality. The strength rank of correlation coefficients was compared with Faizi and Alvi [26]. Correlations between number of cornuti, number of rays and average number of rays per cornutus and specimen were explored for 12 field specimens and 13 museum specimens. Correlations with forewing length and body weight were analyzed in 17 and 15 specimens respectively. These samples had been collected from different locations.

General genital terminology follows Klots [1]. The internal sclerotized plate of the endophallus is called endophallite following Génier [27], but we will retain the more common term cornuti for the star shaped spines of the endophallus. The term endophallus will be used in its morphological sense: the whole integumental structure invaginable in the phallus. The internal duct that runs along the entire phallus, from the ejaculatory duct to the gonopore, will be called the endophallic duct, the term vesica will be used to designate only the eversible part of the endophallus. The retractor muscle of the phallus ($m_6$) is named after Tikhomirov [28].

## Results

### Males produce the cornuti in the genital tract

In the virgin males, the cornuti appear densely packed as a mass approximately centered on the length of the phallic tube. The cornuti are attached to a slightly sclerotized longitudinally folded area (integument) on the right side of the endophalus

(Fig 1A). Opposite to the mass of cornuti, just on the left side of the endophallus, a subtubular well sclerotized plate is present (endophallite). In the everted phalli, both the band with cornuti and the endophallite are always displaced with the rest of the endophallus, but never become completely external (they are non eversible elements) (Fig 1B). The distal portion of the endophallus is distended in such a way that the endophallite becomes rather ventral and the cornuti layer dorsal. The arrangement is such that the rays of the cornuti occupy just the luminal space of the endophallic duct, clearly visible in transversal sections (Fig 1C, 1D).

The cornuti are star-shaped heavily sclerotized structures (Fig 1E) but in section appear not solid (Fig 1F). Through scanning electron microscope, the cornuti appear tightly packed, intimately intertwined (Fig 2A, 2C). The cornuti attached

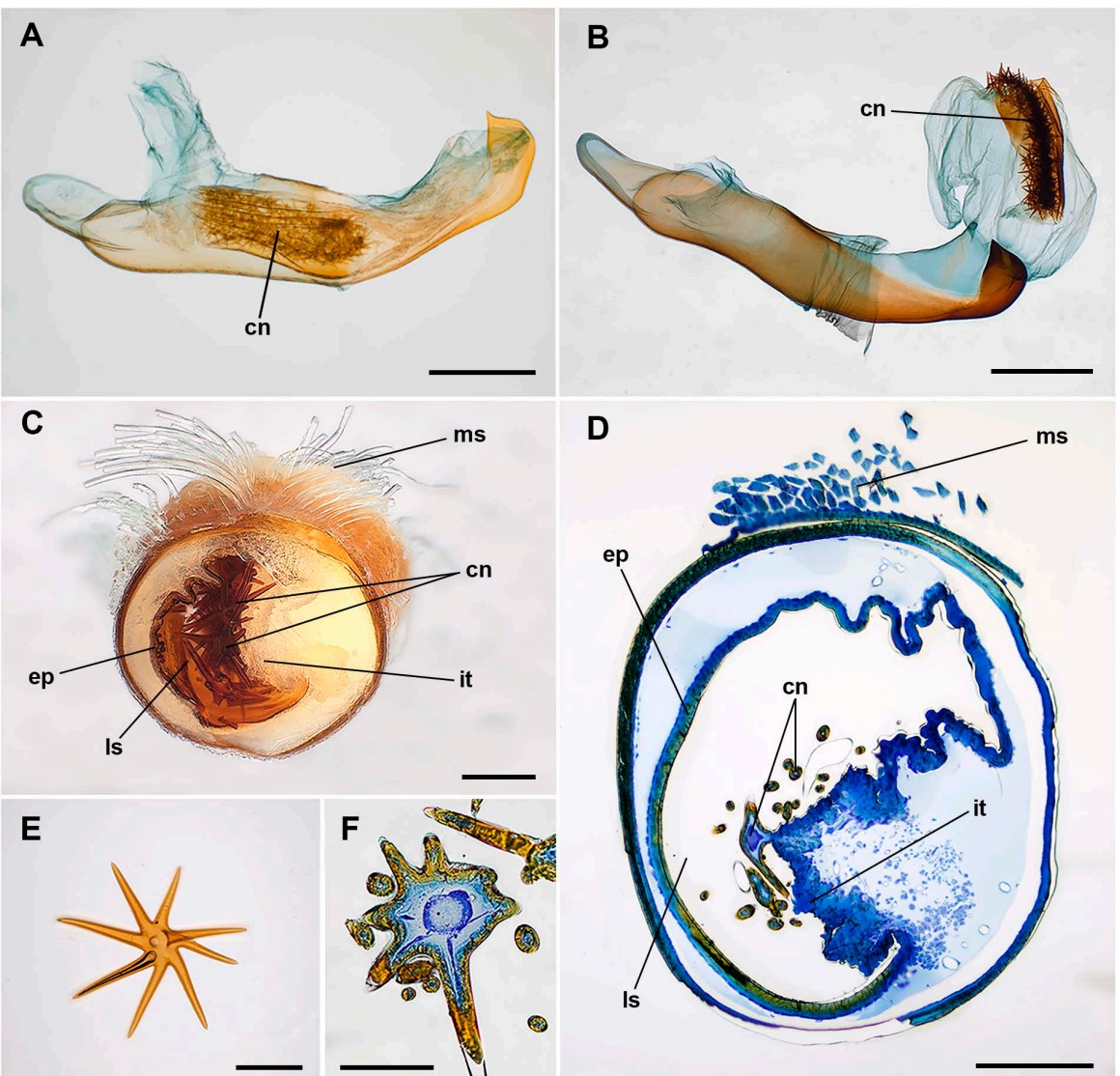

**Fig 1. Phallic tube and cornuti of unmated males of *Peridea anceps*.** A, Non-everted phallic tube, left lateral view. B, Everted phallic tube, left lateral view. C, Cross section of the phallic tube, posterior view. D, Semithin section of the phallic tube, posterior view. E, Single cornutus with eight rays. F, Semithin section of a single cornutus. Abbreviations: cn, cornuti; ep, endophallite; it, internal integument of the vesica supporting the cornuti; ls, luminal space of the vesica; ms, retractor muscles ($m_6$) of the phallus. Scale bars: A, B = 1 mm; C, D, E, F = 100 μm.

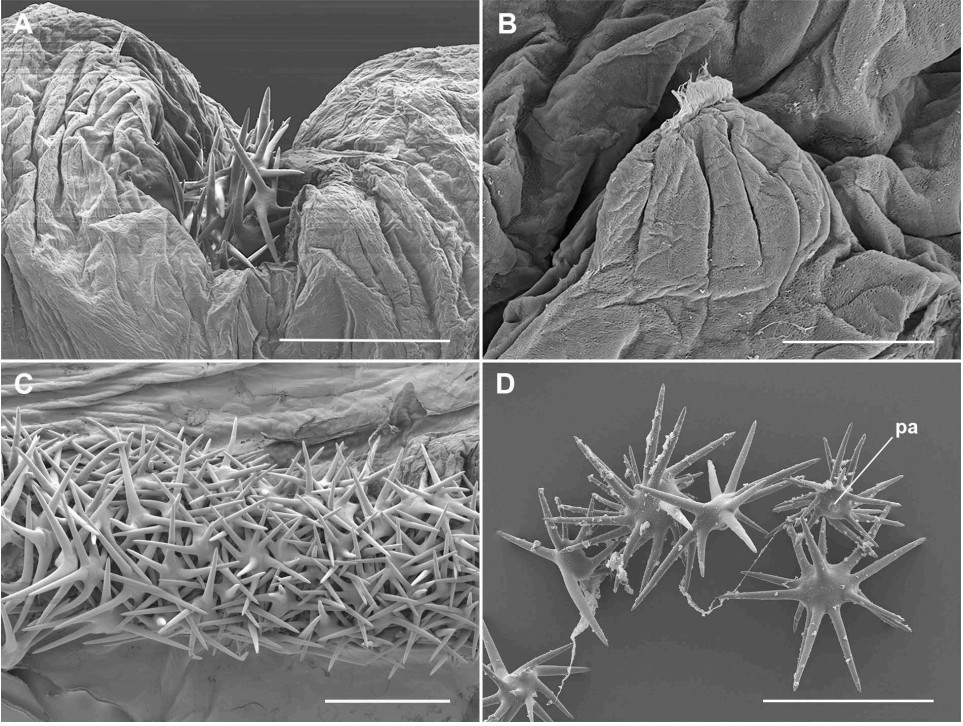

**Fig 2. Scanning electron microscope images of the phallic tube and cornuti of *Peridea anceps*.** A, Male gonopore showing the cornuti; B. Peduncle with cornutus removed; C, Arrangement of cornuti inside the vesica; D, cornuti extracted from the female bursa copulatrix. Scale bars: A, C, D = 300 μm; B = 20 μm. Abbreviations: pa, point of attachment of the cornutus.

at the beginning or at the end of the band are frequently single pointed and, in many cases, do not detach. Each cornutus bears a peduncle for attachment to the integument of the endophallus (Fig 2B). The point of attachment of the cornutus can be identified for detached cornuti (Fig 2D) as a small fragment of endophallus cuticle remains. The peduncle wall is continuous between the heavily sclerotized cuticle of the cornutus and the rather flexible endophallus cuticle. Internally, the peduncle is strangulated and occluded. Consequently, the lumen of the cornuti is not continuous with the endophallus cavity (Fig 3A). Internally, the cornuti are massively vacuolized including membrane whorls (Fig 3B).

The rays of the cornuti are arranged radially from a central body or slightly rotated in a somewhat spiral arrangement (Figs 1E and 2D). The cornuti are variable in number, also variable in the number of rays, length of rays and disposition. Fig 4 includes a sample of the many variations that can be found among cornuti (see also S2 Fig). Tables 1–3 summarize the number of cornuti, rays per cornutus and rays per individual found in the survey.

The average number of cornuti in the whole sample of males was 82 (SD = 14, min–max: 53–120, N = 72) (Table 1). Both the specimens belonging to the group of siblings and the specimens from the field had 85 in average (siblings: SD = 16, min–max: 62–115, N = 15; field: SD = 12, min–max: 62–106, n = 31). The average number of cornuti of the museum specimens was 78 (SD = 16, min–max: 53–120, N = 26).

The number of rays per cornutus ranged from 1 to 24 in the whole sample (Table 2). However, not all the classes were present in each specimen; cornuti with 12 or more rays were poorly represented. The average of rays per cornutus in the whole sample was 6.70 (SD = 1.72, N = 56); the most abundant class was 6 rays. For the sibling specimens, the average was 5.91 (SD = 1.20, N = 31), for the field-collected specimens it was 8.28 (SD = 1.96, N = 12) and lastly, for the museum specimens the average was 7.14 (SD = 1.49, N = 13). The most abundant class for the siblings and museum specimens

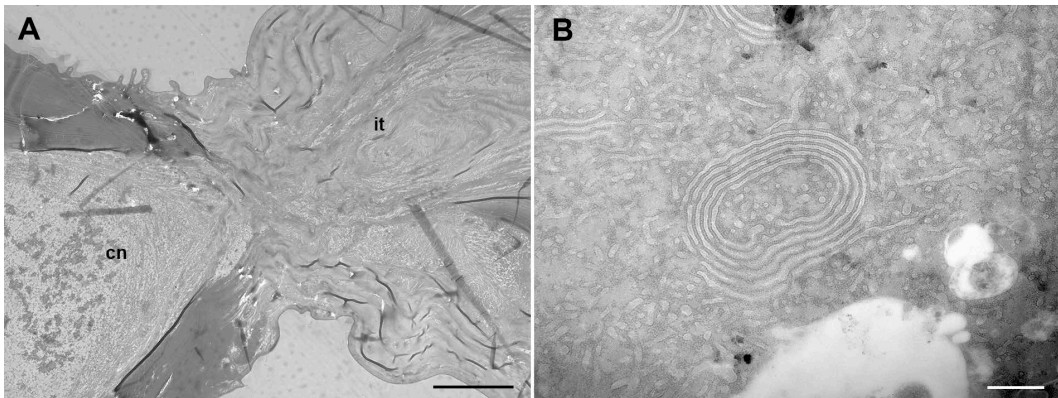

**Fig 3. Transmission electron microscope images of the cornuti.** A, Section of transitional area between the cornutus and the peduncle integument; B, Whorl and other vesicular structures inside the cornutus. Abbreviations: cn, cornuti; it, integument of the peduncle. Scale bars: A = 3 µm; B = 500 nm.

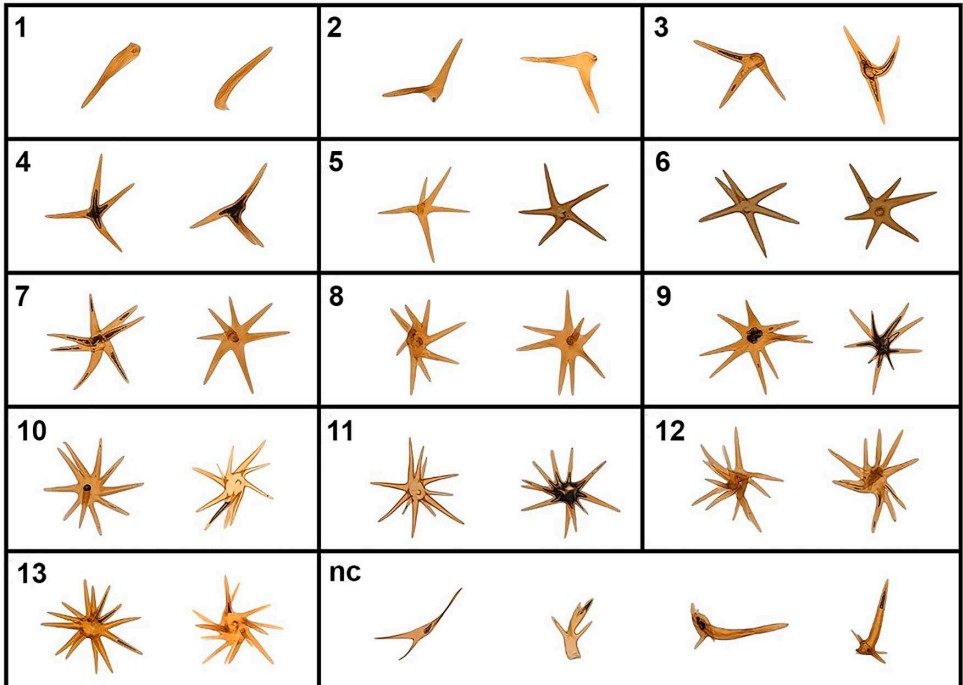

**Fig 4. Examples of cornuti types based on the number of rays.** The number represents the number of rays per cornuti. Abbreviations: nc, uncommon cornuti shapes.

**Table 1. Number of cornuti found in sets of cornuti.**

| Origin of samples | $\bar{x}$ | $M_o$ | $\sigma$ | $Q_{25\%}$–$Q_{75\%}$ | Min–Max | N |
|---|---|---|---|---|---|---|
| Sibling specimens | 85 | 86 | 12 | 80-91 | 62–106 | 31 |
| Field material | 85 | 83 | 16 | 76-93 | 62–115 | 15 |
| Museum collections | 78 | 72 | 16 | 70-88 | 53–120 | 26 |
| Total | 82 | 93 | 14 | 72-91 | 53–120 | 72 |

**Table 2. Number of rays per cornutus found in sets of cornuti.**

| Origin of samples | x̄ | $M_o$ | σ | $Q_{25\%}$–$Q_{75\%}$ | Min–Max | N |
|---|---|---|---|---|---|---|
| Sibling specimens | 5.9 | 6 | 1.2 | 5–7 | 1–13 | 31 |
| Field material | 8.3 | 7 | 1.9 | 6–10 | 1–24 | 12 |
| Museum collections | 7.1 | 6 | 1.5 | 5–8 | 1–17 | 13 |
| Total | 6.7 | 6 | 1.7 | 5–8 | 1–24 | 56 |

was 6 but for the field specimens was 7. In a sample of ten rays (each ray from different cornutus and specimen), the length of the rays (measured from the center of the cornutus) oscillated between 89 and 187 µm (X̄ = 144, SD = 27).

The average number of rays per specimen was 557 (SD = 141, min–max: 299–1026, N = 56) considering the whole sample (Table 3). For the group of siblings, the average number of rays per specimen was 494 (SD = 89, min–max: 299–645, N = 31). For the sample of specimens collected in field, the average of rays per specimen was 705 (SD = 130, min–max: 484–841, N = 12) and for the sample of museum specimens the average was 574 (SD = 153, min–max: 428–1026, N = 13). Globally, out of 117 males dissected 81 (69.2%) had cornuti. In detail, 30 from the siblings (out of 31, 96.8%), 21 from the field-collected specimens (out of 39, 53.8%) and 30 from the museum specimens (out of 47, 63.8%) (S2 Table).

Regression analysis (GLM) did not reveal any significant connection between genital size and the number of cornuti (S3 Table). No significant correlation of number of cornuti with number of rays per specimen was detected; nevertheless with all specimens pooled the correlation is significant (r = 0,30, P = 0,0247). The correlation between the number of cornuti and the average number of rays per cornutus per specimen was significant for all the specimens pooled (P = 0.003) and negative (r = −0.38). It was significant for the siblings (r = −0.51, P = 0.003) and nearly significant for the field males (r = −0.57, P = 0.055) but not for the museum specimens (r = −0.495, P = 0.089). Finally, the correlation analysis between number of cornuti and dry weight and between number of cornuti and wing length did not reveal any significant result (Table SM4). As expected, wing length and body were correlated (r = 0.65, P < 0.001, N = 34).

## Males transfer cornuti to the female genital tract

Dichlorvos-treated live specimens allowed observing the cornuti transfer process that occurs when the spermatophore secretions are injected in the female via the ductus ejaculatorius. The vesica begins to evert and a yellowish liquid material passes through the endophallic duct, at first without detaching cornuti. However, once the vesica is completely everted, a more solid translucid material flows and drags the cornuti allowing the transfer to the female genital tract (S1 File). The drag of cornuti is not always fully efficient and, in some cases, some of the cornuti, either placed at the beginning or at the end of the set of cornuti, remain attached in the endophallus (Fig 1B).

The bursa copulatrix of *P. anceps* is typically a simple sac-like structure. A subspherical corpus bursae is connected to the ostium through a straight ductus bursae. The whole bursa copulatrix is weakly sclerotized except for the presence of a wedge shaped signum in a variably central position of the corpus bursae (Fig 5). The integument is porous. Numerous cornuti were recovered from the bursa copulatrix of the females (X̄ = 71, min–max: 53–88, SD = 12, N = 6). From the whole sample of 19 females collected in the field (including museum specimens), 2 (10.5%) were unmated (with no

**Table 3. Number of rays per specimen found in sets of cornuti.**

| Origin of samples | x̄ | σ | $Q_{25\%}$–$Q_{75\%}$ | Min-Max | N |
|---|---|---|---|---|---|
| Sibling specimens | 494 | 89 | 432–563 | 299–645 | 31 |
| Field material | 705 | 130 | 583–809 | 484–841 | 12 |
| Museum collections | 574 | 153 | 482–577 | 428–1,026 | 13 |
| Total | 557 | 141 | 472–594 | 299–1,026 | 56 |

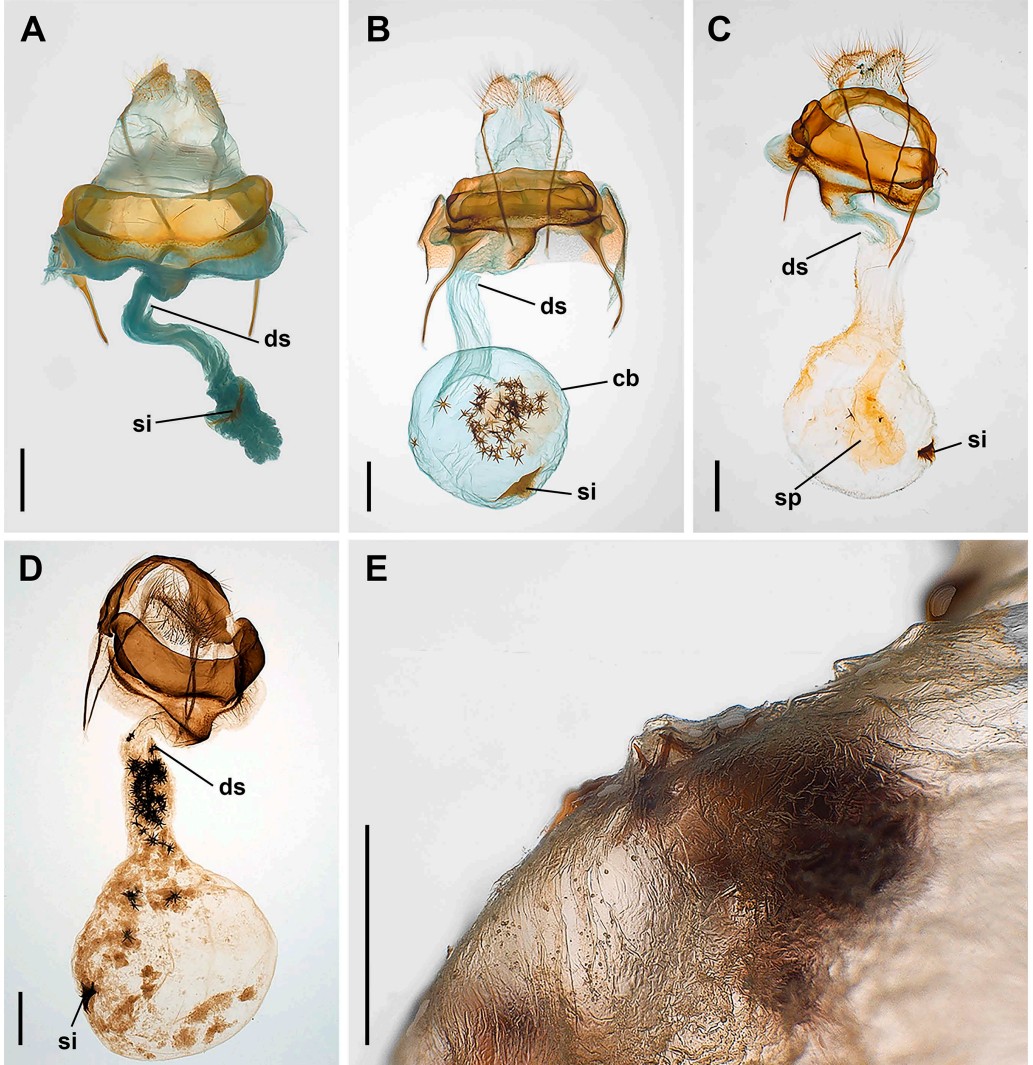

**Fig 5. Female bursa copulatrix of *Peridea anceps*.** A, Virgin female, ventral view; B, Mated female with cornuti inside the corpus bursae, ventral view; C, Mated female with spermatophore and some few cornuti, dorsal view; D, Mated female with cornuti in the ductus bursae, ventral view; E, Detail of a corpus bursae wall with cornuti embedded. Abbreviations: cb, corpus bursae; ds, ductus seminalis; si, signum; sp, spermatophore. Scale bars: A, B, C, D = 1 mm; E = 500 μm.

spermatophore and cornuti), 10 (52.6%) had spermatophore (or remains of) and cornuti, and 7 (36.8%) had spermatophore but no cornuti (S2 Table). From the group of siblings (23 females) only one was mated (with a sibling), containing a single spermatophore and the corresponding set of cornuti (80 cornuti). In the unmated females the corpus bursae appeared contracted, variably folded, without cornuti or sign of spermatophore inside (Fig 5A). In the mated females the corpus bursae was always distended (Fig 5B). When no or few cornuti were detected, a poorly sclerotized piriform spermatophore was found in the female tract (Fig 5C). In two mated females the bursa copulatrix included only few cornuti (1–3). The cornuti, when present, were found in different locations of the bursa copulatrix. They were common in the corpus bursae (Fig 5B), but in six of the ten females with cornuti, they were also abundantly present in the ductus bursae (Fig 5D). Moreover, they could be also found attached to variably degraded fragments of the spermatophore throughout

the bursa copulatrix (Fig 5D). Finally, few cornuti (1–3 in 3 different females) were found loose or even firmly hooked to the corpus bursae wall (deforming it in some cases) (Fig 5E). The observation of cornuti removed from the females by SEM did not reveal any significant microscopic difference from those removed from the males. However, those found in the female tract showed materials adhered to their surface (Fig 2D).

## Discussion

### The cornuti in detail

The cornuti are tightly imbricated in a lateral position of the endophallus. A sclerotized plate, on which the tips of the cornuti cannot attach when sliding outward, facilitating their expulsion and protecting the male from damaging his own endophallus, occupy the opposite side. A membranous plate holds the cornuti, easy to break individually but predictably requiring some force to break collectively (Fig 2A). The neck of the cornuti is blind and its break is mechanical. The cornuti are technically "dead tissue" and the microscopic examination reveals evidence (Fig 3B) of characteristic autophagy [29]. The cornuti must be produced at some point during pupal stage and once formed the connection neck is blocked, driving to death the internal epithelial tissue. Between the cornuti and the sclerotized plate there is a reduced space for the passage of a fluid ejaculate (Fig 1C), suggesting that the least solid materials passes first and the denser material after. The special arrangement of the cornuti indicates that the release of the whole set of cornuti is integral. In fact, no single cornutus could be removed without dragging some of the rest and breaking the whole pack. Moreover, when the cornuti dissections were done it was easier to remove the cornuti with a single movement from the anterior to the posterior part (following the direction of the ejaculates) than on the other way. Incomplete release is rare and when occurs few (less than 10) cornuti remain in the endophallus. It is interesting to note that the presence of these remnants relates to cornuti with few rays, usually one or two rays, which are in the extreme portions of the cornuti plate and not in the central portions.

We consider highly unlikely that males are able to recover cornuti: 1) based on our previous experimental research in another lepidopteran species [11], 2) the general evidence that adult insects lose the ability, present in immature stages, of regenerating organs [30,31], 3) our observations of mated females with spermatophore remains lacking cornuti, and 4) the potential production cost of these structures and the short life span of the adult, among other restrictions, add to the unfeasibility of their recovery.

As far as we know, this is the second quantitative study of intraspecific variation in the number of deciduous cornuti in Lepidoptera [11] and the first regarding caltrop cornuti (but see [9]). Males produce a variable number of cornuti, and each cornutus is different in shape, number, length and disposition of rays. In fact, it could be said that it is difficult if not impossible to find two identical cornuti. No substantial difference in shape or number of cornuti was found that could be attributable to captive breeding that was developed without resource restriction [11]. This study allows us to observe the high variability in the number of rays (Table 1). This variation is observed both within and between males. We can highlight the group of siblings, as it provides insight into the distribution of the number of cornuti and rays in the progeny of a single couple. The number of cornuti contained in the bursa copulatrix of the mother (therefore, the progeny's father) was 88, a number close to the average of cornuti of the male siblings (85).

The morphology of the cornuti deserves some comments. Since Chapman [15] coined the term "caltrop" for the cornuti of the notodontid Lepidoptera, its use has remained unchanged in the otherwise sparse literature on the subject. Strictly, a "caltrop" should respond to a four-pointed (tetrapod) configuration. Although such a configuration is relatively common among the cornuti of *P. anceps*, it is not the most abundant. The number of rays range from 1 to 24, with 5–7 pointed cornuti as the most frequent. Tetrapod designs are widespread in nature and have been subject of study from the physical and engineering point of view in a broad variety of applications (see for example [32–34]). Among the characteristics of these structures, their simplicity, easy packing, ability to interact with other structures and surfaces, and hydrodynamic behaviour stands out; all of them scale-invariant properties. We can expect similar properties in geometries with more

rays, although they imply some sacrifices, the most obvious one being to increase the complexity of their construction. Raising the number of rays can, in the extreme, reduce the ability to interact with each other (making it difficult to mesh) and with surfaces (they could literally roll without sinking in). Therefore, there is a probable constrain on the number of rays, above which the structure begins to lose suitability and below which lose functionality. Although the function of these cornuti is uncertain (see below) they are detached and transferred to the female as a tightly imbricated group. Only later, probably when the spermatophore is being digested, they start to dislodge from each other. Consequently, the whole set probably acts as a functional unit at least for some period after its transfer to the female. The variability observed between cornuti could help tightening the imbrication of the set of cornuti. Muscular movements of the corpus bursae during spermatophore digestion and the ensuing movement of the bursa copulatrix contents could contribute to the later dislodging of individual cornutus from the set (to accomplish additional functions individually?).

Forewing length, body weight and genital size do not explain the number of cornuti. One hypothesis to explain the lack of correlations is that the range of number of cornuti observed corresponds to the range necessary to accomplish their (still unknown) function (see discussion below) and that this "optimal range" is independent of male size or male genitalia size. There is no clear correlation between the number of cornuti and the number of rays per individual, indicating that, with more cornuti, the number of rays is not necessarily higher, and vice versa. Each male will have a unique combination of number of cornuti and number of rays that is not easily predictable. The shape of each cornutus is almost unique, making every set of cornuti distinct: one size does not fit all [35]. The structure is thus a multiple-hook device, without a special reception mechanism for the cornuti in the female, i.e., without a "coupling mechanism" [36].

The correlation between the number of cornuti and the average number of rays per cornutus per individual can be understood as the packing value of the set (see below). Packing would be associated with reducing male production costs, ensuring proper cornuti dislodge and possibly optimal positioning of the structures within the female reproductive tract. As observed, the correlations for two of the three groups of males were significant or marginally significant (the exception was that of museum specimens) and negative. This implies that a higher number of cornuti tends to correspond to a lower average of number of rays per cornutus, although this relationship is not highly consistent. The cornuti caltrop are a complex biological system characterized by high unpredictability, probably aiming to avoid high production costs through packing, resulting in a wide array of combinations that are challenging to replicate.

There is some interspecific variation in number of cornuti, rays and their shape in notodontids [9]. Some characteristics are consequently taxon dependent. However, both the material examined, and the literature (see for example [37]) reveal the cornuti are packed in a similar position and arrangement. We hypothesize that they are transferred with ejaculation inside the female genital tract in a similar way to *P. anceps*. However, there are species closely related to *Peridea* (e.g.,: *Togepteryx* Matsumura, 1920) in the subfamily Notodontinae [38] that do not possess cornuti, suggesting that these genital structures may be lost or gained rapidly, as has been pointed out for Dioptinae [9,17].

## Males transfer the cornuti to the females

Our observations are consistent with the model proposed by Cordero and Miller [9] but add detail. The position and distribution of the cornuti inside the bursa copulatrix together with variable degrees of spermatophore digestion suggest some sequence of events. There is no evidence that females digest or remove the cornuti. The cornuti can obstruct the ductus bursae and cervix (Fig 5D) and accumulate in the corpus bursae (Fig 5E). The cornuti must face the internal cavity of the bursa copulatrix. This internal surface is curved, endowed with mobility and variable.

The cornuti are transferred during the final stages of the spermatophore secretion. Most cornuti are retained primarily at the level of the ductus bursae attached both to the collum of the spermatophore and to the ductus bursae walls. Thus, both the spermatophore and cornuti represent in some way a mating plug immediately after the end of copulation and for some undetermined length of time, probably until spermatophore digestion begins. The movements of the corpus bursae together with the secretion of digestive materials allow access to the nutrients of the spermatophore [7,39]. To what extent

the cornuti contribute to the tearing of the spermatophore envelope and then the digestion of its contents [8] is arguable as the spermatophore walls in *P. anceps* are weak. The bursa copulatrix of the female is muscled and has a well-developed signum whose more common function is breaking the spermatophore envelope [40]. Additionally, the internal vestiture of the bursa copulatrix is rich in pores, suggesting some level of secretion that could also be involved in spermatophore digestion [41]. Moreover, we know some females may be mated without cornuti transference. However, even if the presence of cornuti in the bursa copulatrix does not seem imperative for a correct digestion of the spermatophore, some contribution to the mechanical digestion cannot be ruled out.

As the digestion of the spermatophore progresses, the cornuti descend to deeper parts of the bursa copulatrix allowing contact of the cornuti with the corpus bursae wall. Although the innervation of the bursa copulatrix has not been studied, the movements of the bursa copulatrix could result in the stimulation of mechanoreceptors, such as the stretch receptors found in corpus bursae of the butterfly *Pieris rapae* (Linnaeus, 1758), which are responsible of inducing the mate rejection behaviour of females after receiving a spermatophore [42]. This possible effect of cornuti caltrop on female physiology and behaviour is a fascinating idea whose testing would demonstrate that the cornuti are a peculiar type of secondary socially transferred materials. More vigorous bursa contractions could be prevented in this stage, as they would provoke the cornuti to pierce the walls. The cornuti are observed to sink into the membranous wall of the corpus bursae (Fig 5H). To what extent they can perforate it remains unclear. Our exploration suggests they can stick in the more superficial layers of the cuticle, but no evidence was detected that they can pierce the epithelium.

No female was found with two complete sets of cornuti in the bursa copulatrix. The maximum number of cornuti registered in a female was 88, a number that is better explained by a single mating. Although there is no evidence that *P. anceps* females receive more than one set of cornuti, a previous study of other Notodontidae found evidence of multiple sets in one female of *Scotura nervosa* Schaus, 1896 and two females of *Erbessa priverna* Cramer, 1777 [9]. On the other hand, it is possible to find females with a spermatophore but without cornuti or with residual amounts of cornuti (1–3) (Fig 5D) suggesting that the males may copulate at least twice. Once a set of cornuti has been introduced into the bursa copulatrix the opportunities for a new copulation for the female could be drastically reduced in a species with short-lived adults. Polygyny seems more plausible than polyandry; both, however, infrequent.

## Supporting information

**S1 Fig. Genital measures on the right valva and sclerotized phallic tube.**
(JPG)

**S2 Fig. A set of cornuti extracted from the male.**
(JPG)

**S1 Table. *Peridea anceps* specimens employed for the study.**
(XLSX)

**S2 Table. Status (mated or not) of specimens of *Peridea anceps.***
(XLSX)

**S3 Table. GLM results.**
(XLSX)

**S4 Table. Correlations between caltrop and different variables.**
(XLSX)

**S1 File. Video recording of vesica eversion.**
(MP4)

## Acknowledgments

The authors are indebted to José Luis Yela (Universidad Castilla La Mancha, Toledo, Spain), Mercedes Paris (Museo Nacional de Ciencias Naturales, Madrid, Spain), Michael Falkenberg and Robert Trusch (State Museum of Natural History, Karlsruhe, Germany) and Hossein Rajaei (State Museum of Natural History Stuttgart, Germany) for help with loans of specimens. We would like to thank the staff and equipment of the Electron Microscopy Service of the University of Valencia for their collaboration. Regional and local permissions were provided for collecting and field work.

## Author contributions

**Conceptualization:** Carlos Cordero, Saúl Bernat-Ponce, Joaquín Baixeras.

**Data curation:** Carlos Cordero, Saúl Bernat-Ponce, Joaquín Baixeras.

**Formal analysis:** Carlos Cordero, Saúl Bernat-Ponce, Joaquín Baixeras.

**Funding acquisition:** Joaquín Baixeras.

**Investigation:** Carlos Cordero, Saúl Bernat-Ponce, Jose Vicente Pérez Santa Rita, Joaquín Baixeras.

**Methodology:** Carlos Cordero, Saúl Bernat-Ponce, Jose Vicente Pérez Santa Rita, Joaquín Baixeras.

**Project administration:** Joaquín Baixeras.

**Resources:** Joaquín Baixeras.

**Supervision:** Joaquín Baixeras.

**Visualization:** Joaquín Baixeras.

**Writing – original draft:** Carlos Cordero, Saúl Bernat-Ponce, Jose Vicente Pérez Santa Rita, Joaquín Baixeras.

**Writing – review & editing:** Carlos Cordero, Saúl Bernat-Ponce, Jose Vicente Pérez Santa Rita, Joaquín Baixeras.

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
