## [Decision Letter · Decision Letter 0]

6 May 2025

PONE-D-25-17580The deciduous genital spines in Lepidoptera: peculiar socially transferred materials in the prominent moth Peridea anceps (Goeze, 1781)PLOS ONE

Dear Dr. Cordero,

Thank you for submitting your manuscript to PLOS ONE. After careful consideration, we feel that it has merit but does not fully meet PLOS ONE’s publication criteria as it currently stands. Therefore, we invite you to submit a revised version of the manuscript that addresses the points raised during the review process.

We look forward to receiving your revised manuscript.

Kind regards,

Vazrick Nazari, PhD

Academic Editor

PLOS ONE

Journal Requirements:

“Saúl Bernat was supported by a PhD grant of the Ministerio de Ciencia, Innovación y Universidades (MICIU) of the Spanish Government. Carlos Cordero was supported by a sabbatical grant from PASPA/DGAPA (Universidad Nacional Autónoma de México) and by a grant from the University of Valencia (subprogram “Atracció de Talent”), Spain”

“The first author was supported by a PhD grant of the Ministerio de Ciencia, Innovación y Universidades (MICIU) of the Spanish Government. The participation of Carlos Cordero was supported by a sabbatical grant from PASPA/DGAPA (Universidad Nacional Autónoma de México) and by a grant from the University of Valencia (subprogram “Atracció de Talent”), Spain.”

“Saúl Bernat was supported by a PhD grant of the Ministerio de Ciencia, Innovación y Universidades (MICIU) of the Spanish Government. Carlos Cordero was supported by a sabbatical grant from PASPA/DGAPA (Universidad Nacional Autónoma de México) and by a grant from the University of Valencia (subprogram “Atracció de Talent”), Spain”

Reviewers' comments:

Reviewer's Responses to Questions

**Comments to the Author**

1. Is the manuscript technically sound, and do the data support the conclusions?

Reviewer #1: Yes

Reviewer #2: Yes

Reviewer #3: Partly

2. Has the statistical analysis been performed appropriately and rigorously? 

Reviewer #1: Yes

Reviewer #2: Yes

Reviewer #3: N/A

3. Have the authors made all data underlying the findings in their manuscript fully available?

Reviewer #1: Yes

Reviewer #2: Yes

Reviewer #3: Yes

4. Is the manuscript presented in an intelligible fashion and written in standard English?

Reviewer #1: Yes

Reviewer #2: Yes

Reviewer #3: Yes

5. Review Comments to the Author

Reviewer #1: This is a well polished and very interesting manuscript. My comments are mostly typographical or alternate word choices. There is one sentence in particular that was too ambiguous and should be reworded. Also, it appears that figure 5F is missing.

Reviewer #2: I only have some few critical observations, concerning mostly the Introduction and some Conclusions at the end of the Discussion, see below:

Line 41 “complex structure has evolved rapidly and divergently” – This general statement is taken from a “classical” review, and the speed of changes are only partly supported by more recent papers. I would like to suggest a more moderate formulation here and also in the Abstract. I think, (2) is an old, outdated review! In many groups the genitalia are surprisingly conservative, despite of rapid speciation!

Lines 43-44 “The male intromittent organ, the phallus, is usually a sclerotized simple tubular structure” – the possible (probably) trade-off with the genital capsula should be considered – sophisticated sclerotised phallus vs simplified endophallus and vice versa?!

Lines 48-49 Different origin, morphology and function – it would be necessary to explain here more in details, partly according to (4 and 23: “These modifications of the endophallus are most likely of a different ontogenic origin, perhaps microsculpture or modified setae.”).

Lines 468-469 Important statement which should be indirectly supported by some references on the polyandry in moths with longer and feeding stage of imagoes (e.g. many Noctuids, i.a. pest species).

Reviewer #3: Summarizing the text and content of reviewed manuscript, it is necessary to recommend to the authors:

1) To change the title and bring it into accordance with the content of the work.

2) To reflect in the Abstract the obtained results of the study of the morphology of the cornuti extracted from the aedeagus and vaginal tract of females. To show the revealed correlations as well as their absence between the studied and measured genital structures. To show that leaving the cornuti in the vaginal tract of females does not affect the subsequent copulation of the male with another female and the transfer of the spermatophore to her. To show that not all females have cornuti left in the vaginal tract after copulation. To remove unfounded conclusions.

3) Since the hypotheses expressed by the authors about the functional significance of deciduous cornuti are in fact not confirmed in any way by the obtained results of the study of the morphology of the latter, they can be stated in the discussion and not included in either the Abstract or the Title.

Only a fragment of the review is presented here. The full text of the review is in the attached file.

6. PLOS authors have the option to publish the peer review history of their article (what does this mean? ). If published, this will include your full peer review and any attached files.

**Do you want your identity to be public for this peer review?** For information about this choice, including consent withdrawal, please see our Privacy Policy .

Reviewer #1: No

Reviewer #2: **Yes: ** Zoltán Varga

Reviewer #3: No

---

## [Author Response · Author response to Decision Letter 1]

23 Jun 2025

Answers to Reviewer 1

1) This is a well polished and very interesting manuscript.

Answer: Thank you for your comments.

2) My comments are mostly typographical or alternate word choices.

Answer: All suggestions were accepted.

3) There is one sentence in particular that was too ambiguous and should be reworded.

Answer: We changed that sentence as follows:

Original: “Incomplete release should be considered as a failure more than dosing, as few (less than 10) cornuti remain regularly in the [endo]phallus”

New: “Incomplete release is rare and when occurs few (less than 10) cornuti remain regularly in the [endo]phallus”

4) Also, it appears that figure 5F is missing.

Answer: Yes that was a mistake, the correct figure is 5D.

Answers to Reviewer 2

1) I only have some few critical observations, concerning mostly the Introduction and some Conclusions at the end of the Discussion, see below:

Line 41 “complex structure has evolved rapidly and divergently” – This general statement is taken from a “classical” review, and the speed of changes are only partly supported by more recent papers. I would like to suggest a more moderate formulation here and also in the Abstract. I think, (2) is an old, outdated review! In many groups the genitalia are surprisingly conservative, despite of rapid speciation!

Answer: We partially agree with the reviewer. There are exceptions to this pattern, but most recent reviews (new references 3-5) continue supporting the idea of rapid and divergent evolution. In the revised version we mention that there are exceptions (lines 44-45).

2) Lines 43-44 „The male intromittent organ, the phallus, is usually a sclerotized simple tubular structure” – the possible (probably) trade-off with the genital capsula should be considered – sophisticated sclerotised phallus vs simplified endophallus and vice versa?!

Answer: We were not aware of this pattern (we could not find any reference online). However, although interesting, we think it is not essential for explaining the main results reported in our manuscript.

3) Lines 48-49 Different origin, morphology and function – it would be necessary to explain here more in details, partly according to (4 and 23: “These modifications of the endophallus are most likely of a different ontogenic origin, perhaps microsculpture or modified setae.”).

Answer: We mention and provide a general references regarding the morphological variability of cornuti (line 55). Regarding the function, in the last sentence of this paragraph we cite two references where functional hypotheses are considered (line 56).

4) Lines 468-469 Important statement which should be indirectly supported by some references on the polyandry in moths with longer and feeding stage of imagoes (e.g. many Noctuids, i.a. pest species).

Answer: As for comment 2, we were not aware of this pattern and could not find any reference online. Although interesting, we think it is not essential for explaining the main results reported in our manuscript.

Answers to Reviewer 3.

1) The manuscript is devoted to deciduous genital cornuti in the moths, studied on the example of Peridea anceps (Goeze, 1781). The authors studied the genital cornuti using electron microscopy. The cornuti were extracted from the aedeagus in the males and from the female vaginal tract. The essence of the study itself and its results consist in studying the morphology of genital cornuti, the number and position of their rays. The authors tested the presence of a correlation between the number of cornuti and the number of their rays, between the number of cornuti and dry weight of moth, between the number of cornuti and the length of the moth’s wing, and between the wing length and the moth body. The title of the manuscript, its contents and conclusions require critical comments, which are presented below.

Answer: Descriptive comment. No need to answer.

2) The title of the article.

“The authors discuss the structure and functional significance of the sclerotized armature of the vesica, which has an accepted name, a special term - cornuti. The name will always make it clear to the reader what the article is about. This is the term that should be reflected in the Title of the article. The descriptive name that the authors used in the Title of the article "deciduous genital spines” can mean anything and not necessarily cornuti. For example, numerous spines are located on various structures of the male genitalia - on the valvae, on the hemi-transtila, on the valvellae, they often break off and are lost during copulation. And only the remaining thecae indicate the place where they were located before.”

Answer: The term “spine” is extensively used in the literature to refer to spiky genital structures generally part of the male intromittent organ (but not only, as there may be also in the female), that usually have an interaction with the female tract and that have received lot of attention in the literature on reproductive biology and sexual selection. Certainly, there may be other spines in the genital areas of insects, but the combination “deciduous genital spines” renders unequivocal results in scientific search engines. A search in Google Scholar for the term rendered more than 9,800 results. Most of them are devoted to insects, with a clear preference to Lepidoptera (cornuti) and many of them dealing with the evolutionary role in sexual selection and sexual conflict of spiny genital structures. The same search for “deciduous cornuti” retrieves hardly 900 results, most of them of taxonomic orientation. The reason is that even if “cornuti” is a broadly used term in taxonomic work and has a rather clear meaning in lepidopterological literature, is of limited value when used in a more general context, either morphological or evolutionary. The authors have opted for the publication in Plos One looking for a broader audience. We are convinced that the use of “spines” in the title will attract a broader audience, not necessarily familiar with entomology or Lepidoptera, but also interested in a more general discussion about the form, function and evolutionary analysis of these kinds of structures. The authors do not escape from the use of the term “cornuti” in the more technical side of the manuscript, but consider that in the end, cornuti are spines. Reviewer 3 argues that “spines” are not always “cornuti” and we may accept this, but at the same time does not question that the cornuti are spines. One of the most common forms in which cornuti can appear is that of spines. It is an editorial decision to restrict the scope of the paper or allow a broader audience to the benefit of the publication.

3) “Furthermore, the content of the manuscript does not contain convincing arguments that the falling cornuti are indeed "peculiar socially transferred materials". The authors only express the assumption that they probably are. The inclusion of this assumption in the title of the article, as a substantiated assertion, seems unjustified. In general, based on the content of the manuscript, it is devoted to the study of the structure and construction of cornuti, and the remaining assumptions are at the level of unsubstantiated hypotheses.

Based on the above, the title of the manuscript/article (if it will be accepted and published) could be “The morphology of deciduous cornuti of aedeagus in Peridea anceps (Goeze, 1781) (Lepidoptera: Notodontidae)”.

Answer: The reviewer is right. There is no experimental evidence that the deciduous cornuti transferred by males to females in some Lepidoptera species have an effect on the physiology and behavior of the female that ultimately benefits the male, as required by the definition of “socially transferred materials” (Hakala et al. 2023). For this reason we have explicitly recognized in the title that the role of deciduous cornuti as “socially transferred materials” is hypothetical (“potential” is the term we use in the new title) at this moment. However, we have retained the term in the title for two main reasons. First, because deciduous cornuti, as we have discussed elsewhere (Cordero & Miller 2012; Cordero & Baixeras 2015; Camacho et al 2018), exhibit morphological diversity and, in several cases, complexity (for example, the caltrop cornuti), as well as potentially large production costs (implied by their frequently large numbers) and complex patterns of evolution (several independent origins, lost and regain in several groups, etc.) that are consistent with the idea that selection is involved in their evolution and, thus, that they have an adaptive role at least for males. As Hakala et al. (2023) say referring to “socially transferred materials”: “Often, we have only a poor understanding of the various roles of secondary components or how they work in concert in these complex mixtures. Existing and emerging data sets must be connected with evolutionary theory to maximize the impact of research on socially transferred materials for multiple fields of biology.” Thus, our second reason is that we want to contribute to fill this gap. Caltrop cornuti could represent socially transferred materials and we want to establish this potential link to the evolutionary synthesis in progress proposed by Hakala et al. (2023). In fact, this is the reason why we choose Plos One, a journal aiming at a public interested in wide range of biological problems and their connections. This is also another reason why we do no want to use the term “cornuti” in the title, because we think that it will discourage many potential readers. Taking into account the target readers is one of the facts that need to be taken into account when deciding an effective title for a paper (e.g. Editorial 2021; Pottier et al. 2024).

References:

All papers on deciduous cornuti mentioned in this response are in the literature cited in the manuscript.

Editorial. 2021. Why the title of your paper matters. Nature Human Behaviour 5: 665.

Pottier P. and 14 coauthors. 2024. Title, abstract and keywords: a practical guide to maximize the visibility and impact of academic papers. Proceedings of the Royal Society B 291: 20241222.

4) Abstract.

The authors, having devoted the entire work to describing the structure of cornuti and identifying correlations / or stating their absence between the sizes and number of structures, unexpectedly conclude in the text about the function of deciduous cornute as a plug against subsequent copulation. Although the results obtained do not indicate such a function. The hypothesis could have been stated as an assumption and discussed. However, in the Abstract to the article, the authors already speak in an affirmative form about the function of deciduous cornuti as a confirmed and substantiated fact: “Our results suggest that one of the functions of cornuti in P. anceps is related to protection against sperm competition, although males that have transferred their cornuti are still able to mate and produce another spermatophore.” Although this does not follow from the studies conducted.

Answer: As clearly stated in the text cited by the reviewer, we “suggest” that one of the possible functions of deciduous cornuti is protection against sperm competition. Of course, it is a matter of opinion if our results suggest such function as we think. To attend this criticism, we have modified the abstract and in the last sentence we simply mention that we discuss possible functions of the cornuti caltrop.

And it is even more incorrect to put this unconfirmed conclusion in the title of the article. The statement of another hypothesis in the Abstract is not at all connected with the comparison of statistical measurements: “The possibility that the deciduous caltrop cornuti influence other aspects of the behaviour and physiology of females via mechanical stimulation is a fascinating hypothesis that remains to be tested.” The attempt to link the functional significance of cornuti to mechanical stimulation seems like a somewhat disconcerting parallelism between vertebrates and invertebrates, in this case moths.

Answer: We do not understand why the reviewer said that we included the same “unconfirmed conclusion” (we suppose the reviewer refers to the suggested function) in the title, because in the original title (and in the modified title of the revised manuscript) it was not mentioned. We also don’t understand the criticism regarding the “parallelism” between vertebrates and moths. There is a “classical” paper (cited in the previous and in the revised manuscript) that demonstrates that the corpus bursae is innervated and that its mechanical stimulation with the spermatophore influences the sexual receptivity of females. Thus, mechanical stimulation by the cornuti caltrop is a plausible hypothesis.

5) What are the arguments against the authors' claims?

“If the cornuti performed such important functions as "plugs" limiting repeated mating or mechanical stimulation, then these functions would have been picked up by natural selection and would have been inherent in at least all representatives of the genus. Some closest species of Geometridae differ by presence/ absence of cornuti in aedeagus. “

Answer: Sexually selected characters may evolve rapidly in a phylogenetic context. That is exactly the reason why genitalia may differ between related species. The phylogenetic signal of the character may vary depending on the level of analysis. The more basal genera of Notodontids have deciduous caltrop cornuti, but in some groups have been lost secondarily. The papers by Miller (2009) and Cordero and Miller (2012) (cited in our manuscript) revised this point and we refer the readers to that publication. All the representatives of the genus Peridea, have caltrop cornuti, as many other related genera. We have not studied the case of Geometridae. The adaptive value may be different in different groups.

6) “And according to the results of the study, we see that even in individuals of the same species, during the first copulation, cornuti can remain in the female's vaginal tract, which is not an obstacle to successful mating of the male and the transfer of the spermatophore to the second female. “

Answer: This needs clarification. Virgin males release the cornuti as part of the copulation. As described in the manuscript this must happen by the final steps of the internal interaction, once most of the spermatophore has been released. Cornuti mostly occupy the ductus bursae. Effectively, we may confirm males can repeat a second mating, transferring a new spermatophore. Some few cornuti remaining the male genital tract may then apparently be transferred. However, it is not correct that the cornuti in the female’s vaginal tract are not an obstacle. We have never found a female with cornuti and two spermatophores, suggesting that once the female has received a complete set of cornuti, the female tract is blocked or the digestion of the spermatophore is delayed in such a way that there is no time or opportunity for a new mating.

7) “Moreover, the second female will clearly not receive the corresponding "plug" from the male who has previously lost the cornuti, but this is unlikely to somehow negatively affect the functioning of her reproductive system and egg laying”

Answer: Correct. But the functioning of the female reproductive system is neither affected negatively when she receives the set of cornuti. According to our hypothesis, it is not the goal of the set of cornuti to affect negatively the female system. The goal is to avoid a second rapid access to another male or/and delay the digestion of the spermatophore.

8) “The authors themselves noted that having lost the cornuti, the male is able to mate with the next female. This indicates that the loss of the cornuti during the first copulation is not an adaptive character of the taxon, fixed by natural selection.”

Answer: This is what the evidence suggests. Of course, in a second mating the male wouldn’t have cornuti to transfer and, according to our hypothesis, he would be less effective for preventing/diminishing sperm competition (but not in transferring a spermatophore). A somewhat similar effect is apparently very common in many polyandrous lepidopteran species in which a relatively large sperma

---

## [Decision Letter · Decision Letter 1]

11 Jul 2025

The deciduous genital spines of the moth Peridea anceps (Goeze, 1781): potential socially transferred materials

PONE-D-25-17580R1

Dear Dr. Cordero,

We’re pleased to inform you that your manuscript has been judged scientifically suitable for publication and will be formally accepted for publication once it meets all outstanding technical requirements.

Kind regards,

Vazrick Nazari, PhD

Academic Editor

PLOS ONE

Additional Editor Comments (optional):

Reviewers' comments:

Reviewer 3 recommended accepting the manuscript but also provided some constructive criticism of the responses provided by the authors that I attach here as a separate file.

**Comments to the Author**

1. If the authors have adequately addressed your comments raised in a previous round of review and you feel that this manuscript is now acceptable for publication, you may indicate that here to bypass the “Comments to the Author” section, enter your conflict of interest statement in the “Confidential to Editor” section, and submit your "Accept" recommendation.

Reviewer #1: All comments have been addressed

Reviewer #3: All comments have been addressed

2. Is the manuscript technically sound, and do the data support the conclusions?

Reviewer #1: Yes

Reviewer #3: Partly

3. Has the statistical analysis been performed appropriately and rigorously? 

Reviewer #1: N/A

Reviewer #3: N/A

4. Have the authors made all data underlying the findings in their manuscript fully available?

Reviewer #1: Yes

Reviewer #3: Yes

5. Is the manuscript presented in an intelligible fashion and written in standard English?

Reviewer #1: Yes

Reviewer #3: Yes

6. Review Comments to the Author

Reviewer #1: (No Response)

Reviewer #3: The authors responded to all the reviewer's comments and even made some adjustments to the title, abstract, and text of the manuscript, corrected inaccuracies in morphological terms, and clarified the wording of phrases.

Some deviation from scientific terminology, incorrect interpretation of morphostructures and their homology, ignorance of evolutionary theory and mechanism of the evolutionary process, in particular the process of speciation, vague notions about the concept of species and its criteria will always be obvious to specialists who will give an appropriate assessment of the article. Therefore, I believe that if the authors cannot follow to all recommendations of the reviewer to the detriment of their goals, then with a positive decision of the editors of Plos, the manuscript can be published.

7. PLOS authors have the option to publish the peer review history of their article (what does this mean? ). If published, this will include your full peer review and any attached files.

**Do you want your identity to be public for this peer review?** For information about this choice, including consent withdrawal, please see our Privacy Policy .

Reviewer #1: No

Reviewer #3: No

---

## [Editor Report · Acceptance letter]

PONE-D-25-17580R1

PLOS ONE

Dear Dr. Cordero,

I'm pleased to inform you that your manuscript has been deemed suitable for publication in PLOS ONE. Congratulations! Your manuscript is now being handed over to our production team.

Kind regards,

on behalf of

Dr. Vazrick Nazari

Academic Editor

PLOS ONE